# Psychosocial Factors Related to Stroke Patients’ Rehabilitation Motivation: A Scoping Review and Meta-Analysis Focused on South Korea

**DOI:** 10.3390/healthcare9091211

**Published:** 2021-09-14

**Authors:** Moon Joo Cheong, Yeonseok Kang, Hyung Won Kang

**Affiliations:** 1Rare Diseases Integrative Treatment Research Institute, Wonkwang University, Jangheung Integrative Medical Hospital, 121 Lohas-ro, Anyang-myeon 59338, Korea; sasayayoou@naver.com; 2Department of Medical History, College of Korean Medicine, Wonkwang University, Iksan 54538, Korea; yeonkang@wku.ac.kr; 3Neuroscience Research Center, Department of Neuropsychiatry of Korean Medicine & Inam Neuroscience Research Center, Wonkwang University Sanbon Hospital, Iksan 15865, Korea

**Keywords:** stroke, rehabilitation motivation, psychosocial factors, meta-analysis, scoping review

## Abstract

The incidence of strokes in individuals in their 30–40 s—who have responsibilities towards their families—has increased. Additionally, many stroke patients suffer from post-stroke disabilities and require rehabilitation. However, especially in younger stroke patients, factors such as financial burden and the inability to be productive lead to depression and thereby, the lack of rehabilitation motivation—which affects their therapeutic outcomes. Therefore, medical interventions alone are not sufficient. This study aimed to identify the psychosocial factors that affect stroke patients’ rehabilitation motivation. Hence, a scoping review was conducted to analyze the research trends across South Korean academic papers and theses, followed by a comprehensive meta-analysis to identify the correlations among the variables. Eighteen factors related to rehabilitation motivation were identified. The internal factors were depression, cognition, self-efficacy, self-esteem, disability acceptance, volition, communication, resilience, empowerment, and uncertainty. The external factors included sleep pattern, quality of life, activities of daily living, physical function, social support, financial burden, disease-related characteristics, and rehabilitation environment. Based on these findings, an intervention model should be developed to provide social support to stroke patients. Moreover, psychological interventions should be developed to enhance the self-efficacy of stroke patients who are undergoing rehabilitation.

## 1. Introduction

Stroke is a vascular disease that causes a loss of brain function or dysfunction in the area of the lesion—where the brain cells have died due to insufficient blood supply, resulting from an obstruction or rupture of the brain blood vessels [1]. If a stroke patient does not receive treatment within the golden time, the probability of death increases. Additionally, even if they receive the right treatment at the right time, 70–80% of them experience a post-stroke disability [2]. While strokes are known to occur mostly in older adults, the incidence of strokes in individuals in their 30–40 s has recently increased [3,4]. Compared to older adults, when individuals in their 30–40 s, who have responsibilities as the heads of their households, suffer from strokes, they face a variety of problems [5]. The financial burden from the cost of years of rehabilitation and the inability to perform normal productive activities is a stressor not only on the patient, but also on the family [6]. Moreover, rehabilitation is difficult since stroke patients struggle with depression. Therefore, they feel physically humiliated and emotionally helpless, and have concerns regarding life-long rehabilitation [7]. Above all, successful rehabilitation outcomes depend on the patient’s active participation through their volition and motivation. In this case, rehabilitation is challenged by the patient’s lack of volition towards recovery, even if the therapeutic interventions provided by the specialists and occupational therapists are excellent. Therefore, it is important to explore the psychosocial factors that increase stroke patients’ motivation towards rehabilitation. Furthermore, several recent studies have reported on the importance of psychosocial interventions, rather than using only medical approaches, to enhance stroke patients’ quality of life [8,9]. The Republic of Korea has also been interested in the importance of psychosocial intervention for stroke patients by establishing regional cardiovascular and cerebrovascular disease centers in each region since 2010, as well as implementing support projects.

This government interest has influenced research on psychosocial interventions for stroke patients in various fields such as nursing, counseling, and social welfare. Therefore, it will play an important role in finding effective intervention methods, to systematically explore the psychosocial factors related to the rehabilitation motivation of stroke patients in Korea, announced over the past 10 years [10].

However, there are limited studies that have systematically analyzed the variables that increase stroke patients’ rehabilitation motivation and how psychosocial interventions should be provided [11]. Therefore, it is necessary to systematically organize and synthesize psychosocial intervention papers for stroke patients. Although, there are papers that are excluded due to strict criteria in the case of systematic literature review to synthesize the existing studies.

Currently, it is presumed that there are ongoing discussions on how to increase stroke patients’ rehabilitation motivation. However, although it is crucial that the psychosocial factors related to stroke patients’ rehabilitation motivation should be systematically explored, there are still limitations due to the strict criteria related to the systematic literature review’s methods. Therefore, this study aimed to conduct a scoping review to explore the factors related to stroke patients’ rehabilitation motivation, followed by a meta-analysis of the correlations among the extracted variables related to rehabilitation motivation for a quantitative analysis. Based on these findings, the basic data required to develop psychosocial interventions for stroke patients are provided.

## 2. Methods and Analysis

To understand stroke patients’ rehabilitation motivation related to their therapeutic outcomes, a scoping review was conducted to analyze the research trends across South Korean academic papers and theses, followed by a comprehensive meta-analysis to identify the correlations among the relevant variables.

The following were the scoping review’s procedures:

### 2.1. Stage 1. Identifying the Research Question

The research question should be clearly defined, since the scope of the articles reviewed was determined based on it. The following was this study’s research question: “What are the factors related to South Korean stroke patients’ rehabilitation motivation?”

### 2.2. Stage 2. Identifying Relevant Studies (Information Sources)

This study focused on South Korean articles published between January 2011 and February 2021. The articles comprised academic papers and theses on stroke patients’ rehabilitation motivation. The following national databases were used to search for the articles: Research Information Sharing Service (RISS), Korean Medical Database (KMbase), Korean Association of Medical Journal Editors (KoreaMed), a science and technology infra-service (ScienceON), National Assembly Library (NAL), Oriental Medicine Advanced Searching Integrated System (OASIS), Korean Studies Information Service System (KISS), and Korea Citation Index (KCI). The articles written in English or Korean only were included in the search. The literature search was conducted while considering the limitations related to time and manpower due to the COVID-19 pandemic. Additionally, a specialist and clinical therapist for stroke patients worked as a team to perform the manual search to complete the final set of articles.

#### Search

The search terms comprised the disease (e.g., “stroke”) and intervention (e.g., “rehabilitation motivation or rehabilitation factors related to motivation” OR “self-efficacy” OR “family support” OR “rehabilitation adherence” OR “achievement” OR “psychosocial factors,” including self-motivation, social support, psychological distress, and rehabilitation adherence”).

### 2.3. Stage 3. Study Selection

Two independent researchers, M.J.C. and H.W.K., conducted the study selection according to the scope and recommendations for literature review. After removing the duplicates, we selected and reviewed studies’ titles and abstracts for relevance, as well as evaluated the selected studies’ full texts for eligibility. Any disagreement regarding the study selection was resolved through a discussion. Based on the selection criteria, 52 articles were selected. The selection process was reported according to the Preferred Reporting Items for Systematic Review and Meta-Analysis (PRISMA) Extentions for Scoping Reviews (PRISMA-ScR) guidelines [12] (Figure 1). Targeting articles published between January 2011 and February 2021, a literature search was conducted for 1 month (March 2021) and the results were as follows: A total of 390 articles were collected—99, 88, 20, 20, 35, 22, 25, and 81 articles from ScienceON, NAL, KMbase, KoreaMed, OASIS, KISS, KCI, and RISS, respectively. Among them, 285 duplicated articles were excluded. Of the remaining 105 articles, 54 were excluded after the abstracts were reviewed and found not to be related to stroke patients’ rehabilitation motivation or the research question, or they did not report the factors related to rehabilitation motivation. Consequently, the final set contained 51 articles for the scoping review. Furthermore, 42 articles were included in the correlation meta-analysis for the relevant variables, after excluding nine articles that did not include correlation data.

### 2.4. Stage 4. Data Charting Process

According to the recommendations for scoping reviews, the selected studies’ research theme, similarities, and differences were summarized in a table for data recording. Armstrong et al. [13] recommended that the criteria for scoping reviews be categorized based on the author, publication year, research location, type of intervention, study participants, purpose, methods, result interpretation, and conclusions. Consistent with this, the data charting in this study included the first author’s name, publication year, field, study design, and factors related to rehabilitation motivation. The study participants were limited to stroke patients, research location to South Korea, and study purpose to explorative or effectiveness studies regarding the factors influencing rehabilitation motivation, while no additional recording was made. The data descriptions were as follows. For study design, qualitative and quantitative studies were differentiated. The qualitative studies were further categorized into program effectiveness analysis, survey, and panel investigation. Psychosocial factors related to rehabilitation motivation were categorized into internal (psychological and physical) and external (environmental) factors, while the sub-categories were risk and protective factors. Charting data were recorded using Excel 2016 (Microsoft, Redmond, WA, USA) and were shared among the researchers using Dropbox (Dropbox, Inc., San Francisco, CA, USA) folders. We contacted the selected studies’ corresponding authors via email to request additional information if the data were insufficient or ambiguous.

### 2.5. Synthesis of Scoping Review Results

Data synthesis and analysis were performed using Review Manager (version 5.3; Cochrane, London, UK) and Excel 2016, and the files were shared among the researchers using Dropbox (Dropbox, Inc, San Francisco, CA, USA) folders. Descriptive analyses of the details of the participants, interventions, and outcomes were conducted for all the selected studies. A quantitative synthesis was performed for studies that used the same types of intervention, comparison, and outcome measures. The collected data were analyzed in two stages—first, synthesizing and analyzing the data according to the scoping review process, then classifying the studies with figures that can be meta-analyzed. In the first stage, the scoping review aimed to comprehensively organize and analyze psychosocial variables related to stroke patients’ rehabilitation motivation. Therefore, the studies were classified and coded according to the “author (year of publication),” “participants (patients),” psychosocial factors and sub-factors that affect stroke patients’ rehabilitation motivation, tools used to measure rehabilitation motivation, as well as research methods, procedures, and results. We synthesized and analyzed each article in this manner. In the second step, a meta-analysis of the psychosocial factors related to stroke patients’ rehabilitation motivation was conducted.

### 2.6. Meta-Analysis

After the scoping review, the articles that reported the statistical data of correlations among the variables related to rehabilitation motivation were selected, and the effect size was analyzed for the inter-variable correlations regarding stroke patients’ rehabilitation motivation.

The following were the correlation meta-analysis’ procedures:

#### 2.6.1. Data Coding and Categorization Criteria

To analyze the effect size of the variables related to stroke patients’ rehabilitation motivation, the factors reported in previous studies were first described. Second, each factor was classified as either internal or external factors. Third, the internal and external factors were further classified as either risk or protective factors. Lastly, each factor was classified as psychological, physical or environmental. While coding the data, if there were any disagreements among the researchers regarding the variables’ classification owing to the diversity of the related factors, the opinions of neurological specialists—including this study’s co-author—rehabilitation specialists, and psychologists specialized in motivation were collected to resolve the disagreements. Additionally, to minimize the loss of data regarding the factors related to rehabilitation motivation, multiple factors that shared similar values were merged after this study’s research team and the specialists reached a consensus.

#### 2.6.2. Data Analysis and Effect Size Estimation

To estimate the effect size, test the homogeneity, and analyze the data, MS Excel 2016 was used to code the data, and the Comprehensive Meta-Analysis version 2.0 was used for the meta-analysis. The analysis’ procedures involved testing the publication bias, homogeneity test, analyzing the correlation effect size, homogeneity test based on internal and external factors, analyzing the correlation between each factor’s effect size, analyzing the mediating factors, and conducting regression meta-analysis by year. Additionally, since the correlation coefficients’ distribution was sensitive to the correlation coefficients’ magnitude and the effect size data were converted to Fisher’s Z values for presentation, thus the effect size’s raw data were not reported in this study [14].

### 2.7. Ethics and Dissementation

Ethical approval was not required since the data used in this scoping review did not include individual patient data, and there were no concerns regarding privacy. The results will be disseminated when the manuscript is published in a peer-reviewed journal and/or presented at a relevant conference.

## 3. Results

### 3.1. Synthesizing the Scoping Review’s Results

#### 3.1.1. Year

The number of articles per year was as follows: One article (academic paper) in 2011, six articles (four academic papers and two theses) in 2012, four articles (two academic papers and two theses) in 2013, four articles (four academic papers) in 2014, four articles (three academic papers and one thesis) in 2015, seven articles (four academic papers and three theses) in 2016, ten articles (seven academic papers and three theses) in 2017, five articles (one academic paper and four theses) in 2018, three articles (two academic papers and one thesis) in 2019, six articles (five academic papers and one thesis) in 2020, and one (academic paper) in 2021. Although 2017 had most articles regarding stroke patients’ rehabilitation motivation—ten articles, of which, seven were academic papers—the most number of theses were published in 2018.

#### 3.1.2. Academic Paper and Thesis

Of the 51 articles, 33 (64.7%) were academic papers and 18 (35.2%) were theses. The higher proportion of academic papers may be due to the prioritization of duplicated articles during the frequency analysis. Of the 33 academic papers, 12 (35.2%) were duplicated articles.

#### 3.1.3. Field

The following were the number of articles according to their academic field: 21 (41.1%), 17 (33.3%), seven (13.7%), three (5.8%), two (3.9%), and one (1.9%) from nursing, occupational therapy, physical therapy, psychology and counseling, public health care, and social welfare, respectively.

#### 3.1.4. Study Design and Program

All of the 51 articles were quantitative studies and no qualitative studies were found. This may be due to the fact that questionnaires were commonly used to measure stroke patients’ rehabilitation motivation, since it is a suitable method to explore each variable. Of the 51 articles selected for the scoping review, 29 (56.8%) involved a survey questionnaire and 22 (41.1%) were studies on program effectiveness. The latter were conducted in various ways, while only a small number of studies used the same program. Using a virtual reality program was the most frequent—three articles regarding this were found. The following programs were used across the selected studies: Time-use intervention, task-oriented activity, movie-based or rehabilitation motivation enhancing nursing intervention, discussion of lyrics, game program, stepwise social readjustment, tai-chi exercise, cognitive rehabilitation program, rehabilitation motivation enhancing program, robot-assisted training, virtual reality program, traditional occupational therapy, image training based on mindfulness, pelvic exercise using visual feedback, group art therapy, photographic evidence presentation, and horticultural therapy.

#### 3.1.5. Factors Related to Rehabilitation Motivation

In this study, the following 18 factors related to rehabilitation motivation were identified. The internal factors were depression (17 articles), cognition (two articles), self-efficacy (33 articles), self-esteem (three articles), disability acceptance (two articles), volition (one article), communication (one article), resilience (three articles), empowerment (one article), and uncertainty (one article). The external factors included sleep pattern (two articles), quality of life (five articles), activities of daily living (seven articles), social support (healthcare providers support, family support, peer support, etc.) (seven articles), and financial burden (three articles). The 51 articles that were selected for the scoping review are summarized in Table 1.

### 3.2. Meta-Analysis Result

#### 3.2.1. Publication Bias

The publication bias of the 42 articles that were selected for the final analysis (effect size K = 362) was examined (Figure 2). The funnel plot was used to examine the selected articles’ publication bias, and the Trim and Fill was used as a complementary analysis of the sample bias. As shown in Figure 1, the funnel plot showed an approximate symmetry. Additionally, the sensitivity analysis using Trim and Fill showed that the adjusted value was equivalent to the observed value (*ES_Zr_* = 0.42), as shown in Table 2, indicating the absence of publication bias.

#### 3.2.2. Overall Effect Size of the Variables Related to Rehabilitation Motivation

The overall effect size, based on the 362 effect size values from the 42 selected articles in this study, was estimated as follows (Table 3, Figure 3): The Q value in the homogeneity test at a significance level of 0.05 was 2716.47 (*p* < 0.01). Additionally, H0 was rejected since the data were determined to be heterogeneous. As a result, a random effect model was applied [61]. Moreover, the value of *I^2^*, which indicates the ratio against overall distribution, was 86.71, which is larger than 50, indicating that the level of heterogeneity was considerably high. The results implied that, as each study was conducted using a different method and by a different investigator, the group effect size estimates could not be taken as identical. Therefore, the random effect model was applied in this study, and the result of the analysis showed that, based on 362 correlation K values, the overall correlation effect size was *ES_Zr_* = 0.33, *p* < 0.01, indicating a moderate level [61].

#### 3.2.3. Effect Size of Each Factor Related to Rehabilitation Motivation (Internal, Psychological, Physical, Environmental, Risk or Protective Factor)

The results from analyzing the effect size of the internal, psychological, physical, environmental, risk, and protective factors related to rehabilitation motivation were as follows (Table 4): The effect size was significant for both internal and external factors, *ES_Zr_* = 0.36, *p* < 0.01, and *ES_Zr_* = 0.32, *p* < 0.01, respectively, although no significant difference was found between the internal and external factors (*Q(d)* = 1.73, *p* = 0.19). The effect size was also significant for each of the physical (*ES_Zr_* = 0.31, *p* < 0.01), psychological (*ES_Zr_* = 0.36, *p* < 0.01), and environmental (*ES_Zr_* = 0.32, *p* < 0.01) factors, while no significant difference was found among the three factors (*Q(d)* = 1.96, *p* = 0.38). However, risk and protective factors’ effect sizes were significant, *ES_Zr_* = 0.34, *p* < 0.01; *ES_Zr_* = 0.27, *p* < 0.01, with a significant difference between the two factors (*Q(d)* = 6.41, *p* < 0.05), indicating that the effect size of the risk factors was significantly larger than that of the protective factors.

#### 3.2.4. Analysis of the Internal and External Factors’ Sub-Factors

Among the 362 values of effect size for internal and external factors’ 18 sub-factors, 286 values for seven sub-factors consisting of 10 or more values of effect size for each (across three or more articles) were analyzed in greater detail, as follows (Table 5): Social support which included the family and healthcare provider (*ES_Zr_* = 0.52, *p* < 0.01), self-efficacy (*ES_Zr_* = 0.41, *p* < 0.01), volition (*ES_Zr_* = 0.34, *p* < 0.01), depression (*ES_Zr_* = 0.32, *p* < 0.01), quality of life (*ES_Zr_* = 0.27, *p* < 0.01), and increased activities of daily living (*ES_Zr_* = 0.12, *p* < 0.01), each showed a significant effect size regarding stroke patients’ rehabilitation motivation, with an effect size of 286 sub-factor values (*ES_Zr_* = 0.32, *p* < 0.01). Additionally, the effect size showed a significant difference between each factor (*Q(d)* = 54.79, *p* ≤ 0.01).

## 4. Discussion

This study aimed to explore the psychosocial factors strongly correlated to stroke patients’ rehabilitation motivation. Therefore, a scoping review and correlation meta-analysis were conducted to identify the quantitative data.

First, the scoping review showed that most of the studies regarding stroke patients’ rehabilitation motivation were conducted from 2017 to 2018. This may be attributed to the first Cardiocerebrovascular Disease Management Plan (2018–2022), enacted by the Ministry of Health and Welfare in September 2018 [61]. Notably, stroke patients’ rehabilitation motivation was reported in multiple studies [18,19,20], in addition to Seo et al. [17], which is the key factor. Therefore, the number of relevant studies may have increased around the time the plan was enacted in 2018 [2,3,19]. Additionally, since there were an equal number of academic papers and theses without bias, stroke patients’ rehabilitation motivation is presumed to be a critical factor for both researchers and clinicians. To lend further support, the 52 articles reviewed in this study were conducted across various academic fields, 21 (40.3%), 18 (34.6%), seven (13.4%), three (5%), two (3.8%), and one (1.9%) from nursing, occupational therapy, physical therapy, psychology and counseling, public health care, and social welfare, respectively. However, all of these studies were quantitative in terms of stroke patients’ rehabilitation motivation. Specifically, among the 52 articles, 30 (57.6%) were studies that were conducted using surveys and 22 (42.3%) were studies verifying a program’s efficacy. Only a few studies shared the same program. According to Moon et al. [2], the factors influencing rehabilitation motivation are yet to be adequately explored, which probably drove each researcher to construct individual programs based on arbitrary variables, thus, posing a limitation. This study’s results also showed that of the 52 reviewed articles, depression was studied as a variable in 18 articles. However, the relationship between self-efficacy and rehabilitation motivation was the most frequently (34 articles) investigated.

Second, the correlation meta-analysis to determine the significant correlations between rehabilitation motivation’s factors showed that, for rehabilitation motivation, the internal factors were more significant than the external ones, while the risk factors were more significant than the protective factors for psychological, environmental, and physical variables, in the given order. Hence, to improve stroke patients’ rehabilitation motivation, the focus should be on *self-efficacy* among the internal factors and psychological variables, and on *social support* among the external factors and environmental variables, while a therapeutic intervention should be developed to reduce the risk factors (financial burden, etc.). This was consistent with Kim et al. [24], who suggested that *self-efficacy*, among the motivational factors, enabled people to produce continuous outcomes and participate in therapeutic immersion. Furthermore, *support* as an external factor [26] acts as an external resource. Therefore, as shown in Jung et al. [11], health care providers’ support and family support are crucial in promoting self-mediated rehabilitation in stroke patients. Similarly, Park et al. [23] differentiated the risk factors and protective factors to explore the potential factors that have a key role in increasing or decreasing a given behavior that may affect stroke patients’ rehabilitation motivation through spouses, family, and healthcare providers’ support [62]. Moreover, this study differentiated the risk factors and protective factors in reference to the study by Rodriquez et al. [63] through exploring the factors that may increase or decrease the rehabilitation motivation that contribute to stroke patients’ rehabilitation. Subsequently, reducing the risk factors related to rehabilitation was shown to be more strongly correlated with increased rehabilitation motivation. Monahan and Phipps [64] reported that 70–75% of stroke patients suffer a neurological disorder after the stroke which requires a long-term treatment, while less than 20% of stroke patients are completely cured of the disorder. In this case, it is difficult to completely recover and the possibility of lifelong rehabilitation causes psychological, physical, and financial burdens on stroke patients. Additionally, they may choose to discontinue rehabilitation [32]. Therefore, to meet the demand for rehabilitation for stroke patients in South Korea, specific plans and government support are required to complement regional healthcare services, including medical institutions. Notably, as suggested by this study’s results, it may be necessary for integrated healthcare services to introduce psychosocial interventions for stroke patients and theirs. The following were this study’s significance: First, the psychosocial factors related to stroke patients’ rehabilitation motivation were identified as well as their effects were explored and scientifically verified. Second, the factors’ basic data that can enhance stroke patients’ rehabilitation motivation have been provided for clinicians and specialists who treat stroke patients and provide interventions for them.

The following were the limitations of this study: First, the research method of this study is a scoping review. Therefore, it was not carried out according to the standard of systematic review: PICOS (Participants, Intervention(s), Control (comparators), Outcome(s) and Study Design). Additionally, the research method could not be performed as a qualitative assessment such as risk of bias, which is a procedure of systematic review. Therefore, in future research, we propose to conduct a qualitative assessment such as risk of bias, while collecting and analyzing the studies according to strict criteria such as a systematic review. Second, this study was collected only online. If it was not for the COVID-19 pandemic, it would have been possible to collect gray papers offline. Accordingly, in future research, it is suggested to review studies that have not been published online, including gray papers offline. Third, this paper collected papers related to the rehabilitation motivation of stroke patients, and did not distinguish between acute and chronic stages among stroke patients. The purpose of this study was not to collect the stroke period separately. Therefore, in future research, a comprehensive review of the study is suggested by dividing acute and chronic stages among stroke patients. Lastly, this study was limited to domestic papers. The purpose of this study was to comprehensively review the relationship between psychosocial variables and rehabilitation motivation of stroke patients in Korea, in accordance with the domestic health policy. Therefore, knowing the difference between the psychosocial variables of stroke patients abroad was also limited. In the future, it is suggested that studies synthesize the rehabilitation motivation studies of stroke patients abroad without regional scope.

## 5. Conclusions

This study’s results showed that the factors that had strong correlations with rehabilitation motivation regarding improving therapeutic outcomes for stroke patients were, in order of decreasing effect, social (family and healthcare providers) support, self-efficacy, volition, depression, quality of life, and activities of daily living. Therefore, the support of healthcare providers, family, and friends is predicted, as an external factor, to enhance stroke patients’ rehabilitation motivation, thereby, increasing rehabilitation’s effectiveness. Furthermore, enhancing stroke patients’ self-efficacy as an internal factor of volition and encouragement is anticipated to improve patients’ motivation to engage more actively in rehabilitation.

Based on this study’s findings, the following suggestions were made: First, an intervention model should be developed to provide social support to stroke patients. Second, a psychological intervention method should be developed to enhance the self-efficacy of stroke patients who are undergoing rehabilitation.

## Figures and Tables

**Figure 1 healthcare-09-01211-f001:**
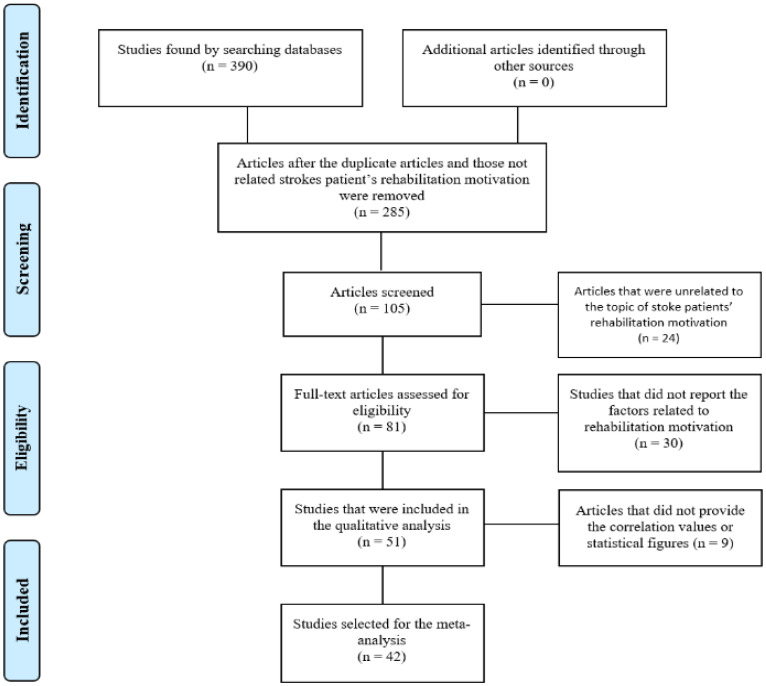
PRISMA Flow diagram.

**Figure 2 healthcare-09-01211-f002:**
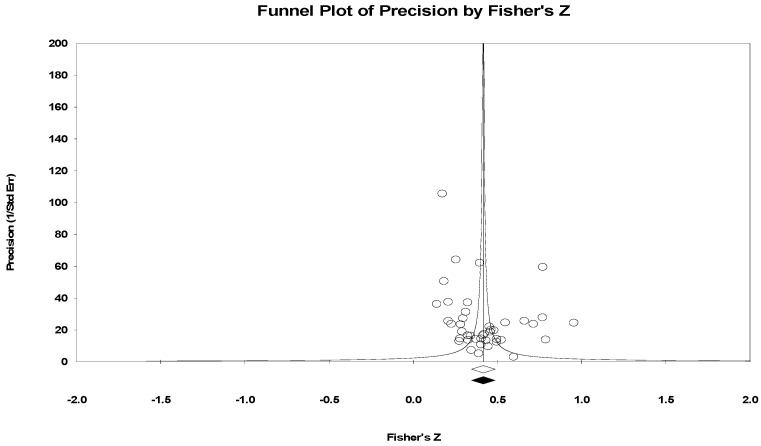
Funnel plot.

**Figure 3 healthcare-09-01211-f003:**
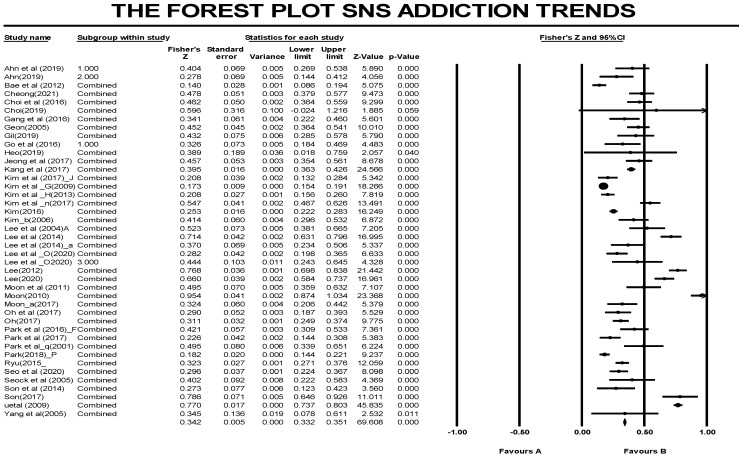
Forest plot.

**Table 1 healthcare-09-01211-t001:** Characteristics of the studies included in the scoping review.

Author (Year)	Article (Thesis)	Field	Study Design (Intervention Methods)	Factors Related to Rehabilitation Motivation
1. Bae et al. (2012) [8]	Article	Occupational Therapy	Survey	Pain catastrophizing, activities of daily living
2. Kim et al. (2013) [15]	Article	Occupational Therapy	Survey	Activities of daily living: ADL self-esteem
3. Lee et al. (2014) [16]	Article	Occupational Therapy	Survey	(Heart type A)/stress
4. Kim et al. (2017) [9]	Article	Occupational Therapy	Survey	Psychosocial factors (depression, stress, self-efficacy), therapeutic flow
5. Seo et al. (2020) [17]	Article	Nursing	Survey	self-efficacy, social support (family support, friends support, healthcare providers support), depression
6. Son et al. (2014) [18]	Article	Nursing/Rehabilitation	Survey	Interpersonal caring behavior, motivation of rehabilitation, QOL, horticultural therapy
7. An et al. (2019) [19]	Article	Nursing	Survey	Self-Esteem, Stroke Specific Quality of Life
8. Moon et al. (2011) [20]	Article	Nursing	Survey	Multidimensional Scale Perceived Social Support
9. Lee et al. (2014) [21]	Article	Nursing	Survey	Self-esteem, rehabilitation motivation/self-efficacy, activities of daily living (ADL)
10. Choi et al. (2016) [22]	Article	Nursing	Survey	Depression, rehabilitation motivation, resilience
11. Park et al. (2016) [23]	Article	Nursing	Survey	Depression, rehabilitation motivation, family support, rehabilitation adherence
12. Kang et al. (2016) [24]	Article	Nursing	Survey	Empowerment, rehabilitation motivation/self-efficacy, perceived stress
13. Choi (2019) [25]	Article	Occupational Therapy	Observational Study (Program effectiveness)	General self-efficacy/rehabilitation motivation
14. Park and Ko (2017) [26]	Article	Nursing	Survey	Rehabilitation adherence, rehabilitation motivation/self-efficacy
15. Kang (2013) [27]	Article	Physical Therapy	Observational Study (Program effectiveness	Rehabilitation motivation/self-efficacy
16. Lee et al. (2020) [28]	Article	Nursing	Survey	Depression, rehabilitation motivation/self-efficacy
17. Kang et al. (2017) [29]	Article	Occupational Therapy	Survey	Sleep quality, stress, rehabilitation motivation
18.Oh et al. (2017) [7]	Article	Nursing	Survey	Depression, rehabilitation motivation/self-efficacy, resilience
19. Go et al. (2016) [30]	Article	Nursing	Survey	Rehabilitation motivation, depression, self-efficacy
20. Jeong et al. (2017) [11]	Article	Nursing	Survey	Social support, uncertainty, rehabilitation motivation/self-efficacy
21. Chung et al. (2012) [31]	Article	Physical Therapy	Survey	Rehabilitation motivation/self-efficacy, gait ability test, stroke specific quality of life
22. Park et al. (2020) [32]	Article	Physical Therapy	Survey	Physical function, balance, cognition, rehabilitation motivation/self-efficacy
23. Song et al. (2012) [33]	Article	Occupational Therapy	Observational Study (Program experiment)	Self-esteem, self-efficacy, family support, rehabilitation motivation, motor function
24. Moon et al. (2018) [34]	Article	Occupational Therapy	Observational Study (Conventional program)	Hand function, modified Barthel index, depression scale, motivation questionnaire
25. Park and Song (2015) [35]	Article	Occupational Therapy	Observational Study (Time-use intervention)	Depression, rehabilitation motivation/self-efficacy
26. Song et al. (2015) [36]	Article	Occupational Therapy	Observational Study (Conventional program, Task-oriented activity)	Rehabilitation motivation, self-efficacy
27. Kwon and Lee (2017) [37]	Article	Nursing	Observational Study (Movie-based nursing intervention)	Rehabilitation motivation/self-efficacy, depression
28. Ca et al. (2020) [38]	Article	Occupational Therapy	Observational Study (Self-check program)	Rehabilitation motivation/self-efficacy, functional recovery
29. Jung (2015) [39]	Article	Occupational Therapy	Observational Study (Discussion of lyrics)	Depression, rehabilitation motivation/self-efficacy
30. Song et al. (2014) [40]	Article	Occupational Therapy	Observational Study (Game program)	Depression, self-efficacy, rehabilitation motivation
31. Park (2017) [41]	Article	Occupational Therapy	Observational Study (Stepwise social readjustment program)	Upper limb function, shoulder pain, functional independence, rehabilitation motivation/self-efficacy, social readjustment,
32. Park et al. (2012) [42]	Article	Occupational Therapy	Observational Study (Tai-chi exercise program)	Upper limb function, functional independence, rehabilitation motivation/self-efficacy
33. Ham (2020) [43]	Article	Occupational Therapy	Observational Study (Cognitive rehabilitation program)	Depression
34. Jeon (2013) [44]	Thesis (Master’s)	Nursing	Observational Study (Rehabilitation motivation enhancing nursing intervention)	Shoulder pain, depression, rehabilitation motivation/self-efficacy, social readjustment
35. Park (2018) [45]	Thesis (Master’s)	Physical Therapy	Observational Study (Robot-assisted virtual reality program)	Rehabilitation motivation/self-efficacy, balance ability
36. Choi (2016) [46]	Thesis (Master’s)	Public health	Observational Study (Traditional occupational therapy vs. virtual reality-based rehabilitation training)	Self-efficacy, rehabilitation motivation
37. Park (2013) [44]	Thesis (Doctoral)	Rehabilitation	Observational Study (Image training based on mindfulness)	Cognition Scale for Older Adults
38. Jung (2019) [47]	Thesis (Master’s)	Physical Therapy	Observational Study (Pelvic exercise using visual feedback)	Balance, gait, activities of daily living, rehabilitation motivation/self-efficacy, quality of life
39. Hyun (2012) [48]	Thesis (Master’s)	Psychological and Counseling	Observational Study (Group art therapy)	Psychological rehabilitation (depression, self-esteem, rehabilitation motivation/self-efficacy)
40. Cho (2016) [49]	Thesis (Master’s)	Physical Therapy	Observational Study (Robot-assisted gait training)	Balance test, gait ability test, self-efficacy and rehabilitation motivation tests
41. GIL (2019) [50]	Thesis (Master’s)	Physical therapy	Survey	Rehabilitation motivation, self-efficacy, functional performance
42. Lee (2020) [51]	Thesis (Master’s)	Nursing	Survey	Depression, social support, self-efficacy, rehabilitation motivation/self-efficacy
43. Moon (2017) [52]	Thesis (Master’s)	Nursing	Survey	Marital intimacy, depression, rehabilitation motivation /self-efficacy
44. Kim (2016) [53]	Thesis (Master’s)	Nursing	Survey	Rehabilitation motivation/self-efficacy, depression, disability acceptance, social support
45. Son (2017) [54]	Thesis (Master’s)	Nursing	Survey	Quality of life, resilience, rehabilitation motivation
46. Oh (2017) [55]	Thesis (Doctoral)	Nursing	Survey	Rehabilitation motivation, activities of daily living, social support (family support, healthcare providers support), depression, self-efficacy
47. Ryu (2015) [56]	Thesis (Master’s)	Occupational Therapy	Observational Study (Program virtual reality)	Depression, rehabilitation motivation/self-efficacy, work participation
48. Lee (2012) [57]	Thesis (Master’s)	Social welfare	Survey	Disability acceptance, social support, rehabilitation motivation/self-efficacy
49. Jeong (2021) [58]	Thesis (Master’s)	Nursing	Survey	Rehabilitation motivation/self-efficacy, healthcare providers support, self-nursing
50. Park (2018) [59]	Thesis (Master’s)	Nursing	Survey	Rehabilitation motivation, risk factor (stroke-related family history, medical history
51.Heo(2018) [60]	Thesis (Master’s)	Psychology	Observational Study (Program effectiveness)	Depression, disability acceptance, rehabilitation motivation/self-efficacy

**Table 2 healthcare-09-01211-t002:** Trim and Fill results.

	Studies Trimmed	*ES_Zr_*	95% CI		Q
			Lower	Upper	
Observed	0	0.42	0.35	0.49	1805.72
Adjusted	0	0.42	0.35	0.49	1805.72

*ES_Zr_*: Correlation effect size; Q: Homogeneity test statistics; CI: Confidence interval.

**Table 3 healthcare-09-01211-t003:** Overall effect size result.

Random Effect Model						Heterogeneity		
Division	K	n	*ES_Zr_*	95% CI		Q (df)	*p*	*I^2^*
				Lower	Upper			
Overall	42	362	0.33	0.30	0.36	2716.47 (361)	0.00	86.71

*ES_Zr_*: Correlation effect size; Q: Homogeneity test statistics; CI: Confidence interval; K: The number of study; n: The number of factors.

**Table 4 healthcare-09-01211-t004:** Effect size of each structural factor related to stroke patients’ rehabilitation motivation.

Structural Factors	N	*ES_Zr_*	95% CI		SE	*p*	*Q(d)*
			Lower	Upper			
Internal factor	94	0.36	0.30	0.41	0.02	0.00	
External factor	264	0.32	0.28	0.35	0.01	0.00	1.73(1)
Physical factor	100	0.31	0.27	0.35	0.02	0.00	*p* = 0.189
Psychological factor	94	0.36	0.31	0.41	0.02	0.00	1.96(2)
Environmental factor	164	0.32	0.28	0.36	0.02	0.00	*p* = 0.375
Risk factor	251	0.34	0.30	0.37	0.02	0.00	6.41(1) *
Protective factor	79	0.27	0.21	0.31	0.02	0.00	*p* = 0.01

*ES_Zr_*: Correlation effect size; Q: Homogeneity test statistics; CI: Confidence interval; N: The number of factors; * *p* < 0.05.

**Table 5 healthcare-09-01211-t005:** Effect size of each sub-factor related to stroke patients’ rehabilitation motivation.

Sub-Factor	N	*ES_Zr_*	95% CI		SE	*p*	*Qb*
			Lower	Upper			
Social support (family support, healthcare providers support, peer support)	32	0.52	0.42	0.61	0.05	0.00	54.79 (5) **
Self-efficacy	16	0.41	0.30	0.52	0.06	0.00	
Volition	42	0.34	0.28	0.40	0.03	0.00	
Depression	49	0.32	0.26	0.38	0.03	0.00	
Quality of life	96	0.27	0.22	0.32	0.03	0.00	
Activities of daily living	21	0.12	0.05	0.20	0.04	0.00	
Overall	286	0.32	0.30	0.35	0.01	0.00	

*ES_Zr_*: Correlation effect size; Q: Homogeneity test statistics; CI: Confidence interval; N: The number of factors; ** *p* < 0.01

## Data Availability

Not applicable.

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
