# Peer review of "Psychosocial Factors Related to Stroke Patients’ Rehabilitation Motivation: A Scoping Review and Meta-Analysis Focused on South Korea"

_healthcare, 2021, doi:10.3390/healthcare9091211_

Round 1

Reviewer 1 Report

Dear authors, 

You have performed an indeed interesting and necessary study on the psychosocial factors affecting stroke patients’ rehabilitation motivation.

I kindly invite you to consider the following aspects:

-Introduction section: However, there are limited studies that have systematically analyzed the variables that increase stroke patients’ rehabilitation motivation and how psychosocial interventions should be provided[10]. Please expand this information.

In general, Introduction section is poorly based on previous literature. I recommend enriching your manuscript in this sense.

-Even having been specified in the manuscript title, I wonder why the review was limited to South Korean academic papers and theses. I strongly recommend widening the scope, so as to gain significance.

-Please carefully check the accuracy of information provided in study design column in Table 1. Sometimes it seems to refer more to intervention than study design.

Thank you and congratulations on your work.

Author Response

Thanks for the nice comments.

The introduction has been reviewed and revised as follows.

[The Republic of Korea has also been interested in the importance of psychosocial intervention for stroke patients by establishing regional cardiovascular and cerebrovascular disease centers in each region since 2010 and implementing support projects.

This government interest has influenced research on psychosocial interventions for stroke patients in various fields such as nursing, counseling, and social welfare. Therefore, It will play an important role in finding effective intervention methods to systematically exploring the psychosocial factors related to the rehabilitation motivation of stroke patients in Korea announced over the past 10 years[10].

However, there are limited studies that have systematically analyzed the variables that increase stroke patients’ rehabilitation motivation and how psychosocial interventions should be provided[11]. Therefore, it is necessary to systematically organize and synthesize psychosocial intervention papers for stroke patients. However, there are papers that are excluded due to strict criteria in the case of systematic literature review to synthesize existing studies.]

And the part where Korean academic papers were reviewed is a limitation of my research. However, this thesis was primarily to explore the current status of stroke patients in Korea. Since then, we have presented an analysis of papers that do not limit the scope to compare domestic and foreign countries. The following is the revised part of this paper.

[Lastly, this study was limited to domestic papers. The purpose of this study was to comprehensively review the relationship between psychosocial variables and rehabilitation motivation of stroke patients in Korea in accordance with the domestic health policy. Therefore, there is a limit to know the difference between the psychosocial variables of stroke patients at abroad.Therefore, it is suggested that future studies synthesize the rehabilitation motivation studies of stroke patients at abroad without regional scope.]

Also, In the case of Table 1, the intervention method described in part of the study design was nomadic and modified as an obervational study.

Once again, thank you for your kind review and thank you for helping me revise.

Reviewer 2 Report

Psychosocial Factors Related to Stroke Patients’ Rehabilitation Motivation: A Scoping Review and Meta-Analysis Focused on South Korea

The authors researched the clinical relevance psychosocial rehabilitation implications of stroke patients in their 30-40s with responsibilities towards their families. These comprehensive meta-analyses of South Korean research identified internal and external factors that influence the rehabilitation process. Based on these findings intervention and psychosocial models have been developed to help young stroke patients undergoing rehabilitation.  

This research is interesting as it sheds some light on factors that influence rehabilitation, its implication and possible resolution.

Overall, the study is fit and only minor English editing in required before publishing.

Author Response

Thank you for your kind and warm review.
English editing and proofreading have been revised once again.
Thank you again.

Reviewer 3 Report

From my point of view, the study design and writing style are good, but it has not challenged a new topic. In other words, the important finding of the study that the need for psychosocial support of the patient and family to increase the motivation for rehabilitation is not a new issue or in other words, it is not a question with an ambiguous answer that requires this study.

Author Response

Thank you for your kind comments.

Regarding the topic, this thesis was written as a problem in the part where the arrangement of interventions on psychosocial variables in stroke patients has not been done quantitatively in the domestic situation.

In particular, it was to confirm the review of the changes in the health policy of stroke patients in Korea since 2010.

The reviewer's opinion is fully understood and considered. In this regard, the limitations were corrected and supplemented as follows.

Lastly, this study was limited to domestic papers. The purpose of this study was to comprehensively review the relationship between psychosocial variables and rehabilitation motivation of stroke patients in Korea in accordance with the domestic health policy. Therefore, there is a limit to know the difference between the psychosocial variables of stroke patients at abroad. Therefore, it is suggested that future studies synthesize the rehabilitation motivation studies of stroke patients at abroad without regional scope.

Thanks again for the great review.

Reviewer 4 Report

This article conducted a scoping review and meta analysis of psychosocial factors related to motivation during post-stroke rehabilitation. The article is limited to studies conducted in South Korea.  Major concerns related to the article include-

  1. Lack of background information on rehabilitation system and insurance policies for rehabilitation in South Korea. Since the study is specifically conducted in a single country, more information should be provided to the readers to help them understand the system.
  2. In the introduction the authors mention that, ‘there are still limitations because of the strict criteria related to systematic literature review’s methods.’- Could you please elaborate on this?
  3. What were the inclusion and exclusion criteria used to select studies?
  4. Did the selected articles include studies conducted during acute stroke and chronic stroke?
  5. Why was the selection of articles limited to the past decade? i.e., between the years 2011 and 2021.
  6. Since the review included both peer-reviewed journal articles and non-peer reviewed thesis- conducting a qualitative analysis to understand the quality of the articles would be helpful.
  7. In methods, the authors state that ‘The literature search was conducted while considering the limitations related to time and manpower due to the COVID-19 pandemic’- could you please elaborate on this?

Author Response

Thank you for your kind and detailed review. The following are corrections to the part you reviewed.

  1. The background we have collected since 2010 is described in the introduction as follows.

[The Republic of Korea has also been interested in the importance of psychosocial intervention for stroke patients by establishing regional cardiovascular and cerebrovascular disease centers in each region since 2010 and implementing support projects.This government interest has influenced research on psychosocial interventions for stroke patients in various fields such as nursing, counseling, and social welfare. Therefore, It will play an important role in finding effective intervention methods to systematically exploring the psychosocial factors related to the rehabilitation motivation of stroke patients in Korea announced over the past 10 years[10].However, there are limited studies that have systematically analyzed the variables that increase stroke patients’ rehabilitation motivation and how psychosocial interventions should be provided[11]. Therefore, it is necessary to systematically organize and synthesize psychosocial intervention papers for stroke patients. However, there are papers that are excluded due to strict criteria in the case of systematic literature review to synthesize existing studies.]

  1. Systematic review should collect, analyze, and evaluate the quality of literature in accordance with the standards of PICOS (participants, Intervention(s),Control(comparator(s), Outcome(s),Study Design). However, as a result of the discussion prior to the collection of this study, studies on stroke rehabilitation motivation and psychological variables in Korea have not been established to follow the PICOS standard. Therefore, we followed the SCOPING REVIEW because we could not follow the strict standards. This part is described in the limit as follows.
  2. Research has described it as follows:

The search terms comprised the disease (e.g., “stroke”) and intervention (e.g., “rehabilitation motivation or rehabilitation factors related to motivation” OR “self-efficacy” OR “family support” OR “rehabilitation adherence” OR “achievement” OR “psychosocial factors,” including self-motivation, social support, psychological distress, and rehabilitation adherence”).

Numbers 4 to 7 are described as the limitations of this study as follows.

The following are the limitations of this study.First, the research method of this study is a scoping review, so it was not carried out according to the standard of systematic review: PICOS(Participants, Intervention(s), Control(comparators), Outcome(s) and Study Desgin).It could not be performed to qualitative assessment such as risk of Bias, which is a procedure of systematic review. Therefore, in future research, we propose to qualitative assessment such as risk of bias while collecting and analyzing the studies according to strict criteria such as systematic review.Second, this study was collected only online. If it wasn't for COVID-19, it would have been possible to collect gray papers offline. Accordingly, in future research, it is suggested to review studies that have not been published online, including gray papers offline. Third, this paper collected papers related to the rehabilitation motivation of stroke patients, and did not distinguish between acute and chronic among stroke patients. The purpose of this study was not to collect the stroke period separately. Therefore, in the future research, it suggests a comprehensive review of the study by dividing acute and chronic stage among stroke patients. Lastly, this study was limited to domestic papers. The purpose of this study was to comprehensively review the relationship between psychosocial variables and rehabilitation motivation of stroke patients in Korea in accordance with the domestic health policy. Therefore, there is a limit to know the difference between the psychosocial variables of stroke patients at abroad.Therefore, it is suggested that future studies synthesize the rehabilitation motivation studies of stroke patients at abroad without regional scope.

Thank you again for your kind and thoughtful review, and thanks to you, we were able to clear up the limitations of our study.

Thank you.

Round 2

Reviewer 3 Report

Dear authors

Thank you for your attention and good explanation.

Reviewer 4 Report

The authors have addressed all comments and concerns previously noted.